# Do Patients with Autoimmune Conditions Have Less Access to Liver Transplantation despite Superior Outcomes?

**DOI:** 10.3390/jpm12071159

**Published:** 2022-07-17

**Authors:** Stephanie S. Keeling, Malcolm F. McDonald, Adrish Anand, Cameron R. Goff, Caroline R. Christmann, Spencer C. Barrett, Michael Kueht, John A. Goss, George Cholankeril, Abbas Rana

**Affiliations:** 1Department of Student Affairs, Baylor College of Medicine, Houston, TX 77030, USA; adrish.anand@bcm.edu (A.A.); crgoff@bcm.edu (C.R.G.); crchrist@bcm.edu (C.R.C.); spencer.barrett2022@gmail.com (S.C.B.); 2Medical Scientist Training Program, Baylor College of Medicine, Houston, TX 77030, USA; malcolm.mcdonald@bcm.edu; 3Transplant Surgery, Department of Surgery, University of Texas Medical Branch, Galveston, TX 77555, USA; mlkueht@utmb.edu; 4Division of Abdominal Transplant, Department of Surgery, Baylor College of Medicine, Houston, TX 77030, USA; jgoss@bcm.edu (J.A.G.); george.cholankeril@bcm.edu (G.C.); abbas.rana@bcm.edu (A.R.)

**Keywords:** autoimmune liver disease, organ allocation, liver transplantation, survival

## Abstract

Orthotopic liver transplantation (OLT) is a lifesaving therapy for patients with irreversible liver damage caused by autoimmune liver diseases (AutoD) including autoimmune hepatitis (AIH), primary biliary cholangitis (PBC), and primary sclerosing cholangitis (PSC). Currently, it is unclear how access to transplantation differs among patients with various etiologies of liver disease. Our aim is to evaluate the likelihood of transplant and the long-term patient and graft survival after OLT for each etiology for transplantation from 2000 to 2021. We conducted a large retrospective study of United Network for Organ Sharing (UNOS) liver transplant patients in five 4-year eras with five cohorts: AutoD (PBC, PSC, AIH cirrhosis), alcohol-related liver disease (ALD), hepatocellular carcinoma (HCC), viral hepatitis, and nonalcoholic steatohepatitis (NASH). We conducted a multivariate analysis for probability of transplant. Intent-to-treat (ITT) analysis was performed to assess the 10-year survival differences for each listing diagnosis while accounting for both waitlist and post-transplant survival. Across all eras, autoimmune conditions had a lower adjusted probability of transplant of 0.92 (0.92, 0.93) compared to ALD 0.97 (0.97, 0.97), HCC 1.08 (1.07, 1.08), viral hepatitis 0.99 (0.99, 0.99), and NASH 0.99 (0.99, 1.00). Patients with AutoD had significantly better post-transplant patient and graft survival than ALD, HCC, viral hepatitis, and NASH in each and across all eras (*p*-values all < 0.001). Patients with AutoD had superior ITT survival (*p*-value < 0.001, log rank test). In addition, the waitlist survival for patients with AutoD compared to other listing diagnoses was improved with the exception of ALD, which showed no significant difference (*p*-value = 0.1056, log rank test). Despite a superior 10-year graft and patient survival in patients transplanted for AutoD, patients with AutoD have a significantly lower probability of receiving a liver transplant compared to those transplanted for HCC, ALD, viral hepatitis, and NASH. Patients with AutoD may benefit from improved liver allocation while maintaining superior waitlist and post-transplant survival. Decreased access in spite of appropriate outcomes for patients poses a significant risk for increased morbidity for patients with AutoD.

## 1. Introduction

Orthotopic liver transplantation (OLT) is a lifesaving therapy for patients with irreversible liver damage caused by autoimmune liver diseases (AutoD) including autoimmune hepatitis (AIH), primary biliary cholangitis (PBC), and primary sclerosing cholangitis (PSC). These patients account for a significant portion of OLTs (24%) in the US and Europe; however, a majority of OLTs are allocated for non-immune etiologies including alcohol-related liver disease (ALD) [1]. In order to maximize the survival benefit of a scarce organ supply, the guiding ethical principles for liver allocation should include *urgency*: the likelihood of dying without a transplant, and *utility*: the likelihood of surviving after transplantation [2]. Currently, it is unclear how access to transplantation differs among patients with various etiologies of liver disease. Understanding these differences may pave the way for future policy changes to reduce disparities in access to liver transplantation.

Both graft and patient survival are excellent in patients transplanted for autoimmune liver disease and have been shown to be superior to those of other transplant cohorts [3,4,5]. Futagawa et al. found that 5-year graft survival of AutoD, PBC (77.3%), PSC (73.3%), and AIH (74.2%) was higher than that of hepatitis B (HBV) (71.5%) and hepatitis C (HCV) (63.2%) [6]. Jain et al. found that 5-year graft survival for AutoD, 73.6%, was higher than the 5-year graft survival rate for alcohol use disorder (64.4%), malignancy (31.8–68%) and over all transplant recipients (59%) [7,8]. Short and long-term patient survival have also been excellent. Afzali et al. report that 1, 3, and 5- year patient survival for AutoD were superior to survival in nonalcoholic steatohepatitis (NASH), ALD, HCV, and HCC [9]. The 10-year patient survival following transplant for AutoD, 62–87%, is superior to that for overall transplant recipients, 57% [3,4,5,6,10,11,12,13,14,15].

While percentages of organs allocated for the various etiologies of liver disease are well-documented, prior studies have not calculated an adjusted probability of transplant for each cohort. This approach may better illuminate the complex factors involved in access to organ transplantation. In addition, prior studies have analyzed post-transplant survival, but intention-to-treat analysis and waitlist survival would provide a better understanding of overall outcomes for various etiologies from the time of listing to 10 years following liver transplantation. The aim of this study is to determine the likelihood of orthotopic liver transplantation as well as the long-term patient and graft survival as a function of the etiology of liver disease from the years 2000 to 2021. To do this, we conducted a large retrospective study of United Network for Organ Sharing (UNOS) liver transplant patients separated into five 4-year eras with five cohorts: autoimmune conditions (PBC, PSC, AIH Cirrhosis), ALD, HCC, viral hepatitis, and NASH. Evaluating the probability of transplantation, waitlist survival, post-transplant survival, and intention-to-treat (ITT) survival after transplant in each cohort will provide a better understanding of OLT outcomes based on etiology of liver disease.

## 2. Materials and Methods

### 2.1. Design/Setting/Population/Procedure

We retrospectively analyzed deidentified patient data of all 218,326 candidates listed for liver transplant between 1 January 2000, and 1 July 2021 using data from the UNOS Scientific Registry of Transplant Recipients (SRTR). Demographic characteristics for patients grouped by era are summarized in Table 1.

### 2.2. Sampling

We exclusively used donor characteristics reported at the time of listing and transplant. The follow-up information was collected at 6 months and then yearly after transplantation. Candidates were followed from the time of listing to death after transplantation or to the last known follow-up. Patients who were lost to follow-up (5.47%) on 1 July 2021 were censored at the date of last known follow-up. Patients were excluded if they were <18 years of age (*n* = 14,468), required a living donor (*n* = 6350), or were waitlisted or received the simultaneous transplant of another organ (*n* = 13,371). Patients who were retransplanted were included (*n* = 10,557).

### 2.3. Eras

Five 4-year cohorts were created to track outcomes over time: 2000–2004 (*n* = 45,786), 2005–2009 (*n* = 49,013), 2010–2014 (*n* = 53,014), 2015–2019 (*n* = 56,663), and 2020–July 2021 (*n* = 18,549). By creating cohorts, we aimed to better assess allocation and survival trends in all listing diagnoses with increased granularity. Demographics for each era can be found in Table 1. Of the patients transplanted, 59.4%, 45.0%, 67.7%, and 85.4% were alive as of 1 July 2021 for each era, respectively.

### 2.4. Etiology of Liver Disease

Patients were categorized into cohorts by the following etiologies of liver disease: ALD, AutoD (PBC, PSC, and autoimmune cirrhosis), HCC, viral hepatitis, and NASH.

### 2.5. Data Collection

UNOS data were analyzed for outcomes for adult liver transplant recipients who received a liver in the 20-year period 2000–2019 using the general-purpose statistical software package Stata^®^ 16.1 (Stata Corp, College Station, TX, USA). Continuous variables were reported as mean ± standard deviation with a *p*-value of <0.05 being considered statistically significant, and all reported *p*-values were 2-sided (*t*-test for normally distributed variables). The primary outcome was defined as overall post-transplant mortality. MELD_PELD_LAB_SCORE was used for the MELD score for each patient.

### 2.6. Data Analysis

Percentage of transplanted patients was calculated as the number of patients transplanted in each era divided by the number of patients listed in that era. Rate of transplantation was calculated by dividing the number of transplants within an era by the cumulative number of years on the waitlist of patients with that diagnosis. Intention-to-treat analysis was performed to assess the 10-year survival differences for each listing diagnosis while including both patients who received a transplant and those who did not but were listed for a transplant. Analysis of waitlist survival was performed to compare outcomes between listing diagnoses for patients who were never transplanted. The multivariate Cox regression analysis results and univariate competing risk between transplant and waitlist death are represented in Cox proportional hazard ratio (HR) with HR > 1 representing increased probability of transplant. For this analysis, all other listings were used as the reference group. Factors used for univariate and multivariate analysis are listed in Table 2. We evaluated every available variable to make the analysis comprehensive for our large study population. Factors significant in univariate Cox regression analysis for increased probability of transplant were included in the multivariate analysis except for listing diagnosis, which was included regardless. The multivariate analysis was conducted to determine the adjusted probability of transplant for a given diagnosis. A hazard ratio above 1 indicated an increased probability of transplant. Post-transplant survival and graft survival for all cohorts were visualized using Kaplan–Meier curves with significance between curves calculated by log-rank test with a Holm correction for multiple comparisons.

## 3. Results

### 3.1. Trends in Transplant Rates and Percentage of Transplantation

The rate at which patients with autoimmune liver disease are receiving transplants is outpaced by that of alcohol-related liver disease in recent years. To assess the dynamics of liver distribution between AutoD, ALD, HCC, viral hepatitis, and NASH over the past two decades, we calculated the percent and rate transplanted for each given condition in different eras (Figure 1). Analysis of the trend for percent of patients transplants for each cohort reveals a consistently higher percentage of HCC patients receiving OLT until the most recent era (2019–2021). The transplant percentage and transplant rate are decreasing for HCC, while increasing for autoimmune conditions, among which ALD, NASH, and viral hepatitis with ALD display the fastest rate.

### 3.2. Multivariate Probability of Transplant

Patients with autoimmune liver disease have a lower adjusted probability of receiving a transplant. To control for confounding variables in liver allocation decisions, we conducted a multivariate Cox regression analysis for the adjusted probability of transplant using the factors in Table 2. The multivariate analysis revealed a significantly lower probability of transplant for patients listed with an autoimmune diagnosis across all eras HR: 0.92 (confidence interval: 0.92, 0.93) compared to ALD 0.97 (0.97, 0.97), HCC 1.08 (1.07, 1.08), viral hepatitis 0.99 (0.99, 0.99), and NASH 0.99 (0.99, 1.00). Despite analyzing the univariate competing risk analysis for probability of transplant versus death on the waitlist, HR for autoimmune conditions remained the lowest among listing conditions (Appendix A).

### 3.3. Intention-to-Treat Survival Analysis

When including all patients with a given listing diagnosis, patients with autoimmune liver disease have a superior survival compared to other etiologies. In addition to understanding the probability of transplant, we sought to determine the survival benefit of transplantation in each condition. To see the effect of patient’s listing condition on mortality, we conducted an intention-to-treat analysis. This survival analysis includes any patent listed for transplant regardless of whether or not they received a transplant. Of all listing diagnoses, patients listed with autoimmune liver disease have improved ITT survival in comparison with ALD, HCC, viral hepatitis, and NASH (Figure 2A, *p*-value < 0.001, log rank test). This includes both patients who received a liver transplant and those who did not.

### 3.4. Waitlist Survival

After excluding those who received a transplant, patients with autoimmune liver disease have waitlist survival equivalent to that of ALD and superior to that of other listing diagnoses. To isolate the specific mortality of patients on the waitlist, survival for patients in the intention-to-treat analysis who did not receive a transplant were compared based on etiology of liver failure. All patients who received a liver transplant were excluded from this analysis. Notably, the 10-year waitlist mortality for autoimmune conditions compared to ALD showed no significant difference (Figure 2B, *p*-value = 0.1056, log rank test).

### 3.5. Post-Transplant Survival

Patients with autoimmune liver disease have superior post-transplant survival. We conducted the same survival analysis in the subgroup of the ITT population for patients that received transplant. Autoimmune conditions had significantly improved post-transplant survival than ALD, HCC, viral hepatitis, and NASH in each era and across all eras (Figure 2C, *p*-values all <0.001, log-rank comparison).

### 3.6. Graft Survival Comparison between Conditions

Patients with autoimmune liver disease have superior graft survival. Finally, we conducted a similar Kaplan–Meier analysis for graft survival in all adults in each condition (Figure 2D). In all adults, autoimmune conditions had significantly better graft survival than ALD, HCC, viral hepatitis, and NASH in each era and across all eras (*p*-value < 0.001, log rank test).

### 3.7. Analysis of Autoimmune Liver Disease

To understand the differences between autoimmune conditions, we conducted Kaplan–Meier analysis in adults with PBS, PSC, and AIH, respectively. PSC had a significantly improved ITT survival compared to PBC and AIH (Appendix A, *p*-value < 0.001, log rank test). PSC also demonstrated a significantly better 10-year post-transplant survival following liver transplant compared to PBC and AIH (Appendix A, *p*-value < 0.001, log rank test). PSC and AIH had significantly better waitlist survival compared to PBS (Appendix A, *p*-value < 0.001, log rank test). Interestingly, PBC showed the highest 10-year graft survival for adults transplanted when compared to PSC and AIH (Appendix A, *p*-value = 0.0038, *p*-value < 0.001 respectively, log-rank test).

## 4. Discussion

This analysis reveals that patients with autoimmune liver disease have superior 10-year post-transplant patient and graft survival, despite having a lower adjusted probability of receiving a transplant, when compared to those transplanted for ALD, HCC, viral hepatitis, and NASH across all eras. This disparity poses a significant risk for increased morbidity for AutoD patients who remain on the waitlist. Our analysis highlights a major concern in allocation policies as alcohol-related liver disease remains the leading etiology of liver transplantation in the United States and trends suggest allocation for ALD will increasingly outpace that of AutoD in the future.

In the MELD-based allocation system in the United States, organs are allocated according to severity of liver disease. This scoring system disadvantages many patients with autoimmune liver disease because MELD score is limited in reflecting the actual severity of liver disease in these patients. For example, the MELD score accounts for kidney disease in its assessment of liver disease. Conditions such as hepatitis C generally have higher MELD scores, due to renal involvement, despite those patients often presenting as clinically more stable than many patients with AutoD who do not have renal disease [16]. Women comprise a large portion of the AutoD population and less of the hepatitis C population; consequently, they are not well represented by the current MELD system [16]. Additionally, women have lower creatinine even during renal dysfunction, which tends to manifest in lower MELD scores [17]. Furthermore, policies regarding exception points favor transplantation for HCC patients, while no such policies exist for other etiologies of liver disease, including AutoD.

Our analysis is consistent with previous literature demonstrating excellent patient survival for patients transplanted with autoimmune etiologies. Prior studies have found 10-year post-transplant patient survivals of PBC (69–87%), AIH (75–80%), and PSC (82–83%) and 5-year post-transplant graft survivals of PBC (77–83%), AIH (75%), and PSC (72–79%) [1,4,10,11,12,13,18,19,20,21]. Long-term graft survival has not been well characterized. Patients with AutoD have consistently higher graft and patient survival compared to other cohorts including hepatitis B, hepatitis C, ALD, and HCC [6,7,8,15].

There remains a discrepancy in demand for liver transplantation in the treatment for AutoD and the probability that these patients will receive the necessary transplant. Medical therapies for chronic cholestatic liver disease have very little effect on disease progression in many patients, leaving liver transplantation as the only definitive therapy for prolonging survival and reversing symptoms of disease [22,23]. While some studies have demonstrated that ursodeoxycholic (UDCA) has a marked impact on clinical outcomes in patients with PBC, up to 40% of patients have an insufficient response and do not see an improvement in survival free of transplantation [24]. In addition, the largest randomized double-blind study by Lindor et al. did not reveal beneficial results with the use of UDCA acid as a treatment for PSC [25]. Although AIH usually responds to immunosuppression therapy, there remains a significant group of patients who develop decompensated liver cirrhosis or fulminant hepatic failure. Liver transplantation is the necessary last resort for those with end-stage liver disease refractory to immunosuppressive therapy [26]. Despite the obvious need for livers in patients with AutoD on the transplant list and their excellent outcomes following transplant, this population is significantly less likely to receive a donor liver according to our study.

Improved survival in patients with autoimmune liver disease is particularly notable given the concern for recurrence of autoimmune disease in allografts. The recurrence of autoimmune conditions has been found in approximately 20–30% of patients after OLT, but it is unclear how this risk changes over time [1,27]. Diagnosis of recurrence remains challenging, as abnormal findings on liver tests may not be specific, and autoimmune markers frequently persist after transplantation [28]. Other complications after transplant such as rejection, graft-versus host disease, bile duct complications, hepatic artery stenosis or thrombosis further complicate the diagnosis as well. In addition, there is variation with regard to biopsy protocols between institutions in the setting of concern for recurrent disease. Due to the difficulties in diagnosing recurrent autoimmune disease and variation in protocols for detecting recurrence, estimates for recurrence are between 0 and 50% in PBC, 12–46% in AIH, and 9–30% in PSC [12,13,19,28]. The short-, medium-, and long-term outcomes of recurrent PBC on graft and patient survival have been negligible in several large review series [3,4,5,6,11,27,29,30].

While recurrence of autoimmune disease is a legitimate concern, clinically significant disease recurrence often responds well to steroids or modification of immunosuppressive regimens in the case of AIH, while recurrent PBC responds to UDCA without jeopardizing patient or graft survival [1,19,31]. Further data on long-term post-OLT outcomes for patients with AutoD may better delineate the clinical significance of disease recurrence on graft and patient survival.

The rate of transplantation for patients with alcohol-related liver disease appears to be increasing dramatically in recent years. A possible explanation of this is changes in recommendations to consider early liver transplant as a salvage therapy with severe alcohol-related hepatitis refractory to steroids. The shift occurred after a landmark case control prospective study challenged the ‘six-month rule’ by showing that the six-month survival for patients with severe alcohol-related hepatitis who did not respond to steroid therapy had significantly improved when compared to those with alcohol-related hepatitis who did not receive a transplant (77% vs. 23%, *p* < 0.001) [32,33]. As more transplant centers no longer require six months of abstinence prior to transplantation, the rate of transplant to patients with ALD will likely continue to increase, which was exacerbated during the COVID-19 pandemic [34]. The novel finding in this study was that waitlist survival does not differ between AutoD and ALD, but overall, patient and graft survival is superior for AutoD.

This analysis raises the question: why do patients who receive transplants for ALD have a significant lower percent survival? This is a major concern in liver allocation, as ALD remains the leading etiology of liver transplantation in the United States, and the rate of transplantation appears to be increasing. Studies have shown that patients with ALD do well in the short term after transplant and have excellent 1- and 3-year survival rates [7,35]. However, they have significantly lower survival rates 5-year post-transplant compared to those with non-alcohol-related liver disease [7]. One explanation is the higher rate of cardiovascular events and de novo extrahepatic neoplasms including lung and upper aerodigestive tract cancer [7]. Alcohol is a known risk for aerodigestive cancers, and lung cancer likely occurs because alcohol and tobacco have a known association. Relapse in alcohol and tobacco use are likely synergistic factors. Nearly 40% of ALD recipients resume smoking and resume it early post-LT [13,36]. Patients who have a relapse of alcohol use show a decreased survival in the long term [2,36].

Rates of relapse range from 10 to 95% likely due to several factors, including variations in the study methodology, the definition and assessment of relapse, and duration of follow-up. In general, between 20 and 50% of the patients who received a liver transplant for end-stage ALD acknowledge some alcohol use in the first 5 year after LT, while 6–17% will resume heavy drinking [36]. Similar reasoning could explain the lower percentage in survival for patients with HCC, as patients with alcohol-induced HCC have worse outcomes compared to non-alcohol induced HCC patients [37].

While prior studies have found similar 1, 3, and 5-year outcomes for patients with NASH when compared to other etiologies, there is also evidence that patient outcomes are inferior in patients with NASH compared to those with AutoD due to cardiovascular and metabolic risk factors [9,38]. Unlike prior studies that have shown some evidence of superior graft survivals in NASH compared to other etiologies, our analysis demonstrated inferior graft survival for NASH compared to AutoD [9,38]. The combination of older age, higher body mass index, diabetes and hypertension common in patients with NASH is associated with post-operative cardiovascular and infectious complications that could explain the inferior long-term outcomes seen in our analysis.

Hepatitis C recurrence following a liver transplant is nearly universal and may lead to progressive allograft injury and failure [39]. Our analysis is consistent with prior studies demonstrating inferior patient and graft outcomes in patients with HCV. However, there is some evidence of equivalent patient and graft outcomes to other etiologies despite disease recurrence [40,41,42]. These studies compared the survival of HCV to that of average overall survival for all etiologies and did not compare to AutoD directly. Our results could potentially be explained by the effect of immunosuppression on viral replication, which may increase the rate and severity of HCV recurrence [38,43,44]. Early HCV recurrence is associated with rapid graft destruction and lower patient survival rates [42]. Recurrence of hepatitis B following liver transplant is less concerning following the advent of immunoprophylaxis with hepatitis B immunoglobulin. With antivirals such as lamivudine or tenofovir alafenamide replacing hepatitis B immunoglobulin in many cases, antiviral-resistant strains of HBV remain a legitimate concern [45,46]. Nonetheless, HCV likely contributes more so to the poor outcomes.

## 5. Limitations

This study is limited by its retrospective designs. Additionally, a large national database registry was used, which is subject to errors and variability; however, given the large size and period of the collected data, small amounts of missing or incorrect data are unlikely to bias analysis in a significant manner. It is even less likely that errors in the data and diagnosis coding were systematically different across each cohort. In addition, since the registry used in this study only records variables at time of listing, time of transplant, and scheduled follow-ups until death, specific data related to clinical events such as rejections, infections, and complications may be unreliable or incomplete.

## 6. Conclusions

This study demonstrates that despite a superior 10-year graft and patient survival in patients transplanted for autoimmune liver disease, these patients have a significantly lower probability of receiving a liver transplant compared to those transplanted for HCC, ALD, viral hepatitis, and NASH. Patients with autoimmune liver disease may benefit from improved liver allocation while maintaining superior waitlist and post-transplant survival. Decreased access in spite of appropriate outcomes for patients poses a significant risk for increased morbidity for patients with autoimmune liver disease.

## Figures and Tables

**Figure 1 jpm-12-01159-f001:**
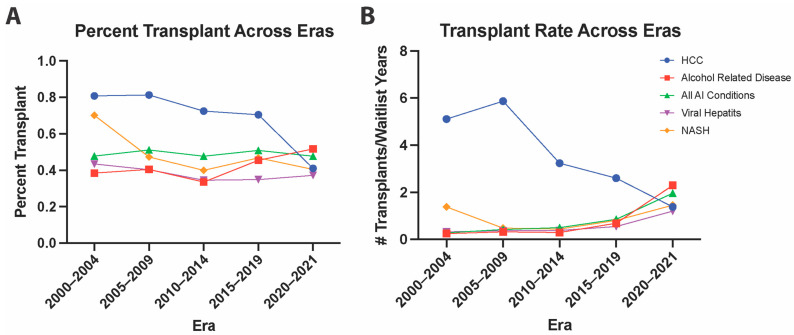
The 20-Year Trend in Transplantation for HCC, Alcoholic Cirrhosis, and Autoimmune Conditions, NASH, and Viral Hepatitis. The percent transplanted (**A**) or transplant rate (**B**) of patients listed with either HCC, Alcohol-Related Disease, Autoimmune Conditions, NASH, or Viral Hepatitis who were transplanted were graphed across each five-year era.

**Figure 2 jpm-12-01159-f002:**
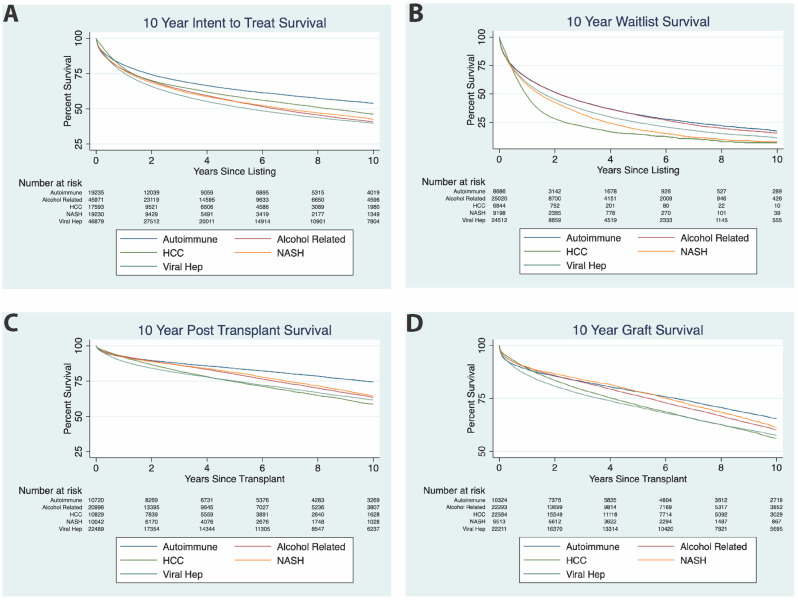
ITT Survival, Waitlist Survival, Post-Transplant Survival, and Graft Survival for Liver Transplantation HCC, ALD, Autoimmune Conditions, Viral Hepatitis, and NASH. (**A**) ITT Survival, all AutoD curves are significantly different from other curves (*p*-value < 0.001, log rank test). (**B**) Waitlist Survival, all AutoD curves are significantly different from other curves (*p*-value < 0.001, log rank test) with exception of ALD (*p*-value = 0.1056, log rank test). (**C**) Post-transplant survival, all AutoD curves are significantly different from other curve (*p*-value < 0.001, log rank test). (**D**) Graft survival, all AutoD curves are significantly different from each other (*p*-value < 0.001, log rank test, NASH (*p* = 0.0345, log rank test).

**Table 1 jpm-12-01159-t001:** Donor and Recipient Patient Demographics.

Table 1: Patient Demographics						
	2000–2004	2005–2009	2010–2014	2015–2019	2020 July 2021	Total
Number of patients	44,390	48,037	51,940	55,060	18,549	217,976
Listing age, mean (SD)	51.1 (10.1)	52.9 (10.2)	55.0 (10.2)	55.7 (11.0)	54.9 (11.8)	53.9 (10.6)
Gender (%F)	36.72	35	35.12	36.41	37.79	35.97
Race (%)						
White	73.44	71.28	70.33	69.8	70.15	71
Black	7.87	8.62	9.05	7.8	7.08	8.23
Hispanic	13.4	14.33	14.77	16.2	16.93	14.94
Asian	4.18	4.67	4.47	4.44	4.02	4.41
% Transplanted	48.7	45.8	49.5	44.6	48.87	52.2
Years waitlist, mean (SD)	1.45 (2.79)	1.09 (2.25)	0.913 (1.78)	0.587 (1.08)	0.251 (0.384)	0.927 (1.99)
Diagnosis (% All Patients)						
PBC	3.75	2.88	2.54	2.49	2.51	3.05
PSC	4.67	4.13	3.92	3.83	4.03	4.28
Autoimmune Cirrhosis	3.32	2.91	2.96	2.97	2.75	3.12
HCC	4.62	12.19	17.67	17.06	12.77	14.7
Alcoholic Cirrhosis	21.95	20.87	21.15	28.29	36.05	25.63
Viral Hepatitis	37.38	34.06	31.68	16.97	8	27.64
NASH	1	6.16	10.78	17.92	18.03	10.17
Donor Characteristics						
Mean cold ischemia time hours (SD)	7.76 (3.59)	7.18 (3.31)	6.35 (2.58)	5.94 (2.13)	6.35 (2.97)	6.67 (2.95)
DRI (SD)	1.67 (0.38)	1.69 (0.40)	1.66 (0.37)	1.68 (0.40)	1.78 (0.43)	1.68 (0.39)
DCD (% Donors)	1.21	2.78	2.82	4.49	4.13	3.02

**Table 2 jpm-12-01159-t002:** Listing Variables Used for Multivariate Analysis.

Table 2: Listing Variables Used for Multivariate Analysis		
African American	Encephalopathy	Payment Method
Age	Highest Level of Education	Private
18–30	High School Dropout	Medicaid
60–65	High School	Region
>65	Technical	Serum Na
Albumin	Bachelors	<125
2.0–2.5	Doctor	125–130
1.5–2.0	ICU	130–135
<1.5	INR	145–150
Ascites at Listing	<2.5	150–155
Bilirubin	2.5–3	>155
<2	3–3.5	TIPSS
8–16	3.5–4	Transplant Location
16–32	>4	Regional
>32	Life Support	National
BMI	MELD	Foreign
30–35	30–35	Ventilator Status
35–40	35–40	Working
>40	>40	

## Data Availability

The data that support the findings of this study are available in the Scientific Registry of Transplant Recipients (SRTR) database which can be requested from SRTR at www.srtr.org (accessed on 31 May 2022).

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
