# Peer review of "Do Patients with Autoimmune Conditions Have Less Access to Liver Transplantation despite Superior Outcomes?"

_jpm, 2022, doi:10.3390/jpm12071159_

Round 1

Reviewer 1 Report

This article is interesting. But the conclusions still require more evidence, or changning the presentation of results.  Questions:

1.      Better post-transplant prognosis cannot support the conclusion that AutoD patients deserve more graft allocation,because AutoD patients also have higher survival on waiting list. Authors should analyze the survival difference between receiving transplant and not to show the real benefits.

2.      Figure2A, B and Supplemental Figure 1A, B are about survival, but the figure legends are about mortality. The two statements should be unified.

3.      Why AutoD patients have a lower probability of liver transplant in the US?Foreign readers may need more background information.

Author Response

Dear Reviewer for the Journal of Personalized Medicine,

   Thank you so much for your consideration of our manuscript. We greatly appreciate the time and effort of the reviewers for providing opportunities for us to improve our manuscript. Below are the point-by-point responses to the comments from Reviewer 1.

  1. Better post-transplant prognosis cannot support the conclusion that AutoD patients deserve more graft allocation because AutoD patients also have higher survival on waiting list. Authors should analyze the survival difference between receiving transplant and not to show the real benefits.

After reconsideration, we recognize that this is a legitimate concern with the analysis.

 Our intent to treat analysis includes those who receive a transplant and those who do not.  It reveals improved ITT survival for those with autoimmune liver disease (Page 6, Line 227). This survival benefit is directly related to receiving a transplant. Our analysis also demonstrates equivalent waitlist survival for AutoD compared to ALD, but superior to that of other etiologies (Page 7, Line 303). Because of this concern, we have toned down our language and removed mention of changes in allocation policies. However, we maintain that this is an important disparity to recognize and poses a significant morbidity risk for those with autoimmune liver disease.

  1. Figure2A, B and Supplemental Figure 1A, B are about survival, but the figure legends are about mortality. The two statements should be unified.

We agree and have unified the statements to reflect survival.

  1. Why AutoD patients have a lower probability of liver transplant in the US?Foreign readers may need more background information.

We agree that further clarification is needed and have added a paragraph referencing the use and limitation of the MELD score for allocation purposes in the United States (Page 8, Lines 434).

  1. Improve results and conclusion

We agree that the results and conclusion portion of the manuscript needed improvement. We attempted to make findings in the results more organized and clearer. We changed the language in the conclusion to better reflect findings in our analysis without overreaching with the suggestion policy changes.

Reviewer 2 Report

Nice paper that statistiscally represent what we have always thought about the patient in the autoimmune condition. I would like to sujest to modify the scheme in the supplement table 1 and add in the title that is a cox regression and add in the table that you are evaluating the Hazard Ratio.

Author Response

Dear Reviewer for the Journal of Personalized Medicine,

   Thank you so much for your consideration of our manuscript. We greatly appreciate the time and effort of the reviewers for providing opportunities for us to improve our manuscript. Below are the point-by-point responses to the comments from Reviewer 2.

  1. I would like to suggest to modify the scheme in the supplement table 1 and add in the title that is a cox regression and add in the table that you are evaluating the Hazard Ratio.

We agree that this increases clarity of supplemental table 1 and have modified it accordingly.

Round 2

Reviewer 1 Report

The authors' explanations about the intercomparison of AutoD and ALD are reliable. I would still like to see an analysis of survival data for patients on the waiting list who do not end up receiving a liver transplant to see if patients on the waiting list with AutoD have less severe disease. However, the data for AutoD and ALD are sufficient to suggest that allocation policies are likely unreasonable between AutoD and ALD.

In addition, the survival graphs in the article are not labeled reasonably, and "10 year survival" is redundant, suggesting ITT, waitrlist survival, etc.

Author Response

  1. The authors' explanations about the intercomparison of AutoD and ALD are reliable. I would still like to see an analysis of survival data for patients on the waiting list who do not end up receiving a liver transplant to see if patients on the waiting list with AutoD have less severe disease. However, the data for AutoD and ALD are sufficient to suggest that allocation policies are likely unreasonable between AutoD and ALD.

We apologize for the confusing wording and have clarified the specifics of our analysis (Page 4, Line 175; Page 7, Line 324). Our waitlist analysis removes all patients who received a transplant and demonstrates equivalent waitlist survival for AutoD compared to ALD, but superior to that of other etiologies (Page 7, Line 326).

  1. In addition, the survival graphs in the article are not labeled reasonably, and "10 year survival" is redundant, suggesting ITT, waitrlist survival, etc.

We agree and have changed labels for Supplemental Figure 1.